# Observational Study to Compare Biological Drug Concentration Quantification Techniques and Immunogenicity in Patients with Immune-Mediated Diseases

**DOI:** 10.3390/biomedicines12040839

**Published:** 2024-04-10

**Authors:** Alejandro Martínez-Pradeda, Laida Elberdín, Ángeles Porta-Sánchez, María Outeda, Mª Teresa Diz-Lois Palomares, Teresa Vázquez-Rey, Benito González-Conde, Emilio Estévez-Prieto, María I. Gómez-Besteiro, Isabel Martín-Herranz

**Affiliations:** 1Department of Pharmacy, Complexo Hospitalario Universitario de A Coruña (CHUAC), Sergas, Instituto de Investigación Biomédica de A Coruña (INIBIC), Universidade da Coruña (UDC), 15006 A Coruña, Spain; laida.elberdin.pazos@sergas.es (L.E.); angeles.porta.sanchez@sergas.es (Á.P.-S.); isabel.martin.herranz@sergas.es (I.M.-H.); 2Department of Gastroenterology, Complexo Hospitalario Universitario de A Coruña (CHUAC), Sergas, Instituto de Investigación Biomédica de A Coruña (INIBIC), Universidade da Coruña (UDC), 15006 A Coruña, Spain; mt.diz.lois.polamares@sergas.es (M.T.D.-L.P.); teresa.vazquez.rey@sergas.es (T.V.-R.); benito.gonzalez.conde@sergas.es (B.G.-C.); emilio.estevez.prieto@sergas.es (E.E.-P.); 3Clinical Epidemiology and Biostatistics Unit, Complexo Hospitalario Universitario de A Coruña (CHUAC), Sergas, Instituto de Investigación Biomédica de A Coruña (INIBIC), Universidade da Coruña (UDC), 15006 A Coruña, Spain

**Keywords:** therapeutic drug monitoring, TDM, ELISA, CLIA, anti-drug antibodies, biological drugs

## Abstract

Measuring biological drugs’ trough concentrations and the concentrations of anti-drug antibodies is a valuable practice for treatment optimization. ELISA techniques are the gold standard for biological drug concentration quantification, but new techniques such as chemiluminescence immunoassays present some advantages. The aim of this unicentric prospective observational study is to compare the infliximab, adalimumab, vedolizumab and ustekinumab trough levels and anti-adalimumab and anti-infliximab antibodies concentrations obtained when using a chemiluminescent instrument (i-TRACK^®^, Theradiag, Croissy-Beaubourg, France) and an ELISA instrument (TRITURUS^®^, Griffols, Barcelona, Spain). Linear regression, Pearson or Spearman tests, Bland–Altman plots and the Cohen kappa test were applied for every sample. The correlation was excellent for both assays in the measurement of all drug concentrations. In general, values were lower when measured using i-TRACK than when using TRITURUS, especially when the values were high. Both techniques proved valuable in clinical practice for monitoring adalimumab and infliximab drug concentration. However, the results were modest for ustekinumab and vedolizumab, so caution is recommended and further research is needed. The limited number of anti-drug antibody-positive samples precluded a comparison between the techniques.

## 1. Introduction

Biological agents have brought about a new era in the therapeutics of immune-mediated inflammatory (IMIDs) diseases (such as inflammatory bowel disease (IBD), rheumatoid arthritis (RAA) and ankylosing spondylitis (AS)).

Therapeutic drug monitoring (TDM) for biologic agents such as infliximab, adalimumab, vedolizumab and ustekinumab has been shown to improve disease outcomes in many IMIDS, specially IBD [1,2,3]. Measuring anti-tumor necrosis factor alpha (TNFa) drug trough concentrations and the concentrations of anti-drug antibodies (ADAs) is a valuable practice for treatment optimization, and it is also cost-effective. In addition, infliximab TDM has been proven to be a strong predictor of the likelihood of achieving mucous healing in IBD after dose intensification [4]. Nowadays, TDM for biologic agents is recommended by various international guidelines [5,6,7,8,9,10].

Several methods have been described for biologic agent TDM, including the Enzyme-Linked ImmunoSorbent Assay (ELISA), radioimmunoassays, chemiluminescence immunoassays (CLIA), and homogeneous mobility shift assays based on liquid chromotography or liquid chromotography coupled with mass spectrometry. ELISA assays remain the most commonly used method in clinical practice and in research, and caution must be taken when applying the therapeutic window established for one drug’s trough level using assays other than ELISA [11,12].

However, ELISA techniques present some limitations. First, they should be applied to a series of samples in order to make the process more cost-efficient. Secondly, trained laboratory staff are needed to perform these assays. Finally, require a relatively large amount of time to obtain results (more than 2 h). Therefore, these assays do not allow practitioners to make immediate dose adjustments [13,14].

Recently, some new assays have been commercialized for TDM, such as the CLIA. CLIA is an immunoassay that has several advantages over conventional ELISA. Firstly, it works with single samples, so there is no need to pool samples. Secondly, it allows the process to be automated, and therefore standardized. In addition, its duration is shorter, approximately half an hour. Therefore, CLIA could be a better tool for immediate decision making by clinicians [15].

However, little is known about the quality of the correlation of the measurements that it provides with efficacy and safety in clinical practice. Only a few studies have compared these new techniques with the gold standard, in some cases with a reduced sample size [16,17,18,19].

The aim of this study was to compare the infliximab (cINF), adalimumab (cADA), vedolizumab (cVED) and ustekinumab (cUST) trough levels and ADA concentrations obtained using CLIA (i-TRACK^®^, Palex, Theradiag, Croissy-Beaubourg, France) and ELISA (TRITURUS^®^, Griffols, Barcelona, Spain) analyzers in patients diagnosed with IMID.

The selection of target drugs for this study (infliximab, adalimumab, vedolizumab and ustekinumab) was based on their widespread clinical use, as supported by international clinical practice guidelines for the treatment of immune-mediated diseases. These therapeutic agents have consistently demonstrated efficacy in managing various immune-mediated conditions, and therapies employing these agents represent cornerstone therapeutic options in clinical practice. Importantly, in some patients, treatment decisions heavily rely on TDM, further underscoring the relevance of this study in the clinical context [1,9,10]. The inclusion of these drugs reflects their prominence in clinical management protocols, emphasizing the potential benefits of employing rapid drug concentration and ADA measurement methods, such as CLIA, particularly for these extensively used therapies, facilitating timely treatment decisions.

## 2. Materials and Methods

This was a prospective, descriptive, observational study.

### 2.1. Study Population

Patients with IMIDs treated with adalimumab, infliximab, ustekinumab or vedolizumab by the Department of Digestive of University Hospital of A Coruña (Spain) from December 2021 to June 2022 were included. Patients were candidates for inclusion in this study if they were at least 18 years old and were treated with adalimumab, infliximab, ustekinumab or vedolizumab for at least 12 weeks. Both patients receiving biosimilar medicines and original brand medicines were included.

Demographic data (age and sex), anthropometric data (weight and height), treatment-related data (drug, dosage and starting date) and drug sample data (date of extraction, time since the administration of the last dose, drug plasma concentration and ADA plasma concentration measured by the different analytical techniques) were collected.

### 2.2. Samples Collection

The serum samples for the measurement of biologic drug and ADA concentrations were drawn immediately prior to drug administration in order to obtain the drug trough level. The blood samples were collected in a tube with a separator gel for serum extraction. The same sample was used for the two different assays. The samples were stored at between 2 °C and 8 °C for a maximum of 48 h after being collected. For longer storage periods, the samples were aliquoted and frozen (at between −20 °C and −80 °C).

Sample collection was performed only from Monday to Friday, so in those patients whose next dose of drug coincided with the weekend, the extraction was performed on a Friday.

Due to the aim of this study, when more than one sample was collected from the same patient at different times, they were considered as different samples.

### 2.3. Measurement of Biologic Drug Serum Levels

Drug and ADA concentrations were determined in each sample. The assay procedures were performed according to the manufacturer’s instructions and the specific protocol for each instrument.

Drug concentrations were expressed in micrograms per milliliter (μg/mL). ADA concentrations were expressed in nanograms per milliliter (ng/mL) in the case of i-TRACK and in international units per milliliter (IU/mL) in the case of TRITURUS. It was not possible to equalize these units of measurement because the commercial house does not provide the equivalence of international units with mass measurements.

The TRITURUS device establishes the following lower limits of quantification (LLOQs): 0.01 μg/mL for cINF, 2 IU/mL cADA-INF, 0.01 μg/mL for cADA, 6 IU/mL for cADA-ADA, 0.63 μg/mL cUST and for 0.8 μg/mL cVED. On the other hand, the i-TRACK device sets the following LLOQs: 0.3 μg/mL for cINF, 10 ng/mL for cADA-INF, 24 μg/mL for cADA, 10 ng/mL for cADA-ADA, 0.1 μg/mL for cUST and 1 μg/mL cVED [20]. The upper limits of quantification were suppressed by performing dilution in those samples in which it was necessary.

The technical sheets of both assays establish that the measurements are not affected by the presence and quantity of other analytes such as serum bilirubin or triglycerides [20,21,22].

### 2.4. Statistical Analysis

All measurements were carried out by the same laboratory, so it was not necessary to perform imprecision control tests.

Values lower than the limit of quantification were expressed by the assays as categorical values (<0.01 or <0.5). From a clinical point of view, these values indicating the lack of a quantifiable concentration of the analyte (either a drug or ADA) require individual assessment through the integration of drug and ADA concentration data. The presence of low values could be attributed to issues of sensitivity in the analytical method, such as the identification of endogenous antibodies unrelated to the specific analyte under consideration [10,15]. When conducting the statistical analysis, two preliminary analyses were conducted to account for values below the limit of quantification (LOQ). In the first approach, these values were considered as quantitative variables equal to “0”. In the second approach, they were deemed as “missing” and excluded from the analysis. No discrepancies were observed in the outcomes of the analysis using either method. Consequently, it was determined that treating these data as “missing” was more suitable in order to prevent potential biases in estimation arising from assuming non-actual “0” values.

Descriptive analysis was performed on the quantitative variables expressed as the mean ± deviation, median and range. For the qualitative variables, frequency and percentage analysis was carried out with estimation of the 95% confidence intervals.

To study the correlation of the values according to the two techniques used, linear regression was performed, and the Pearson or Spearman correlation index was calculated as appropriate after determining using the Kolmogorov–Smirnov test whether or not the variables were normally distributed.

To evaluate the agreement between the two techniques through their quantitative values, as well as to establish the limits and visualize outliers, Bland–Altman graphs were created and the average overestimation was calculated. To evaluate the agreement with the data stratified into categories, the Kappa index was calculated.

### 2.5. Ethical Considerations

The Ethics Committee for Clinical Investigation of Galicia (Spain) approved this study (Protocol Code 2021/352). All patients provided written informed consent before inclusion in this study. This study was conducted in accordance with the Helsinki Declaration of 1964 and its later amendments.

## 3. Results

### 3.1. Samples Characteristics

A total of 188 samples were included: 79 for adalimumab, 23 for infliximab, 76 for ustekinumab and 10 for vedolizumab. After removing the values that were categorical because they did not reach the LOQ, the number of samples for cADA, cINF, cUST, cVED, cADA-ADA and cADA-INF was 75, 17, 76, 10, 4 and 6, respectively.

### 3.2. Data Correlation

Linear regression and Spearman’s or Pearson’s r are shown in Figure 1.

In the case of drug concentration determinations, Spearman’s and Pearson’s test values showed an excellent linear correlation, from 0.93 to 0.98. This correlation was much lower in the case of the cADA-ADA concentration, although the cADA-INF correlation was higher.

### 3.3. Data Agreement and Biases

#### 3.3.1. Adalimumab

Regarding the results of the Bland–Altman plot (Figure 2), for cADA, the mean difference was −2.38 (95% CI −12.71 ± 7.95), with a standard deviation of of 5.27. Therefore, the mean of the differences was quite close to 0, which indicates that both methods produce similar results. The values were aggregated within the lower range (under 10 μg/mL). However, as the average of the two measurements increased, the difference started to increase, being on average 18.3% lower using i-TRACK. In addition, five values were above the upper limit of agreement, each with a mean value of more than 25 μg/mL, meaning that 93.3% of values were within the agreement limits.

#### 3.3.2. Infliximab

In the case of infliximab, the results obtained indicate a mean difference of 0.29 (95% CI –2.65 ± 3.22), with a standard deviation of 1.50. As in the previous case, the mean difference is very close to 0, indicating a similar measurement for both methods. Differentiating from cADA, only two values are aberrant, with one higher than the limit of agreement and the other lower (88.2% values between the agreement limits). In any case, the rest of the values remained aggregated as the average of the values increases, with a 4% overestimation for i-TRACK, and the bias was 0.29.

#### 3.3.3. Ustekinumab

The Bland–Altman plot for cUST presents greater disparity than the previous drugs. The mean difference was −1.46 (95% CI −4.25 ± 1.32), with a standard deviation of −1.47. However, when the average of the two measurements increased, the difference started to increase, being on average 36.7% lower when using i-TRACK than when using TRITURUS. Furthermore, when cUST values were high (greater than 7 μg/mL), a large disparity occured, with four aberrant values (94.7% within the limit of agreement).

#### 3.3.4. Vedolizumab

Finally, in the case of vedolizumab, the calculated mean difference was −12.77 (95% CI −30.30 ± 4.75), much higher than the value of 0. The standard deviation was 8.944. Values from i-TRACK were 38.7% lower than TRITURUS. No value was aberrant, although the limits of agreement presented a very wide range.

#### 3.3.5. ADA Concentrations

ADAs were only detected in four adalimumab samples and six infliximab samples, so it is not possible to compare the results between both techniques.

### 3.4. Qualitative Analysis

In the qualitative analysis, the results for each technique were divided into three categories in the case of cADA, cINF and cUST: a subtherapeutic range, an optimal range and a high range. In the case of cVED, the results were divided in two categories (a subtherapeutic and optimal range). All of the categories were established according to worldwide guidelines and expert recommendations [23,24,25]. Kappa values are shown in Table 1.

Following the recommendations of Landis et al. [26], the strength of agreement obtained for cADA and cINF was “almost perfect” (kappa > 0.81) and “moderate” for cVED and cUST (kappa > 0.41), respectively.

## 4. Discussion

In this study, we compared two techniques for quantifying concentrations of biologic drugs, TRITURUS (ELISA test) and i-TRACK (CLIA test), in patients with immune-mediated inflammatory diseases (IMIDs) receiving adalimumab, infliximab, vedolizumab, and ustekinumab. CLIA belongs to a group of TDM trials that allow for rapid results and immediate clinical decisions, so it is necessary to evaluate whether its results are comparable with the gold-standard technique.

### 4.1. Drug Concentrations

In general, the results obtained when comparing the agreement of both methods differed substantially depending on the drug studied.

First, in the case of cADA, the correlation was very good, and it seems that, in general, there was a similar measurement between methods. However, the Bland–Altman plots show that there was a tendency for i-TRACK to underestimate cADA values, especially with higher cADA values. This underestimation occurs when cADA values are higher than the upper limit of the optimal range. In addition, the qualitative analysis demonstrated high agreement between TRITURUS and i-TRACK, with only a few samples for which the interpretation of the value would be different. Therefore, from a clinical point of view, this underestimation may not be relevant, except in those cases in which dose reductions or other strategies that require certainty about the concentration in a high range are sought.

Two other studies have evaluated CLIA techniques as well. In both cases, the overall results are similar, although one of them found a tendency for i-TRACK to overestimate the results [27], while the other one found an underestimation, like in our results [15]. Variations in assay sensitivity, specificity and calibration standards could contribute to these differences. Differences in sample processing, assay reagents and lots and detection methods may also play a role. Additionally, factors such as assay interference, matrix effects and antibody cross-reactivity could affect the accuracy of drug concentration measurements [1,2,17]. Further investigations into these methodological differences are essential to elucidate the underlying reasons for these discrepancies.

The results from cINF analysis are also consistent. The correlation and agreement were very good. There were no under- or overestimations in cINF values, except in extremely high values. The qualitative analysis was “almost perfect”. Although our results indicate that there was a very good agreement and correlation between both techniques, it is important to highlight that the sample size is small, and furthermore, the proportion of patients with concentrations below the therapeutic range is very low. Given that precisely these patients with subtherapeutic concentrations are the most interesting for decision making, our results should be treated with caution. Nevertheless, another study with a larger sample size found similar results to our study [15], while another one found slightly worse results, although it used a more restrictive therapeutic range [27].

The results obtained for cUST and cVED are not as good as the previous results.

On the one hand, in the case of ustekinumab, the correlation was almost perfect and the Bland–Altman plots show a small mean difference in absolute terms (a bias of 1.46). In addition, the techniques demonstrated “moderate concordance”, assessed using the Kappa index, which means that a significant portion of samples would be interpreted differently in practice. From a clinical point of view, this can be decisive, because the therapeutic range of the drug (according to the literature, between 1 and 4.5 μg/mL) [28] is a very narrow range, so small deviations or imprecisions would be very influential.

Our study analyzes a significant number of samples, so there could be more limitations when using this technique. It is likely that this type of technique is not the most suitable for drugs that display such low concentrations.

To the best of our knowledge, this is the first study that compares CLIA with ELISA tests for ustekinumab. Other authors have studied other techniques such as the drug-tolerant high mobility shift assay (HMSA), but they found that agreement was poor between the HMSA and two ELISA tests [29], while electrochemiluminescent immunoassays (ECLIA) were found to display good correlation and agreement [30], although these authors provided limited data about the statistics used.

On the other hand, in the case of vedolizumab, “moderate concordance” was observed using the Kappa index. The correlation was almost perfect, but the Bland–Altman plots reveal a tendency for i-TRACK to underestimate the cVED values as the concentrations increased, quantified as 38.5% lower on average, in addition to a mean difference line much higher than 0.

It is true that in the case of cVED, the therapeutic range established by the literature (>25 μg/mL) [31] is a relatively wide range, which reduces the impact of these deviations if we compare them with those of the other drugs.

The small number of samples analyzed could justify these inferior results. It is noteworthy that among the 10 samples obtained, while TRITURUS would place 50% within the therapeutic range, i-TRACK could only place 30% within the therapeutic range, with the important implications in clinical practice that this would have. It is obvious that the sample size greatly reduces the applicability of these findings, but caution should be taken when using this technique until more studies are carried out.

To date, we have not found any research about cVED quantification techniques other than ELISA. There are a few studies that compare different techniques to determine antibodies [32] or different ELISA tests [33], but in our study, vedolizumab ADAs were not determined. This could be due to the less experience that exists in the TDM of this drug compared to the others described in our work.

### 4.2. ADAs Concentration

The limited number of ADA-positive samples did not allow a comparison between the techniques. In addition, in the analysis of antibodies against drugs (ADA and INF), each technique used different measurement units, and it was not possible to obtain equivalence. This is a limitation of every study about ADA quantification, because no international analytical standard is currently available, as was explained in the international guidelines [9].

In recent years, the American Gastroenterological Association has established that it is reasonable to study the correlation between anti-drug antibody assays, but it is not as ideal to perform quantitative studies, because the results must be interpreted in the appropriate clinical context and with the drug concentration levels [9]. Although the numerical results have discrepancies, overall it appears that the assays provided similar guidance for clinical practice in the majority of patients with a loss of response [34].

Other authors have recommended that practitioners should not interchange tests for drug and ADAs, especially in one individual’s follow-up [35], which is supported by our findings.

Consistent with prior research assessing techniques for quantifying biologic drugs, our findings align with studies demonstrating a high correlation and concordance among different methods. These investigations consistently underscore the importance of selecting the most appropriate technique tailored to each specific biologic drug [15,24,27]. Notably, our study provides further evidence supporting the integration of CLIA into clinical practice. The expeditious results offered by CLIA, coupled with the comparable performance with ELISA in our research, advocate for its adoption as a valuable tool for routine monitoring in patient care. This highlights the potential of CLIA to enhance the efficiency and precision of therapeutic drug monitoring, potentially optimizing treatment outcomes for patients with IMIDs. Moving forward, larger-scale studies are needed to validate the utility of CLIA and refine its application in therapeutic drug monitoring protocols.

### 4.3. Limitations

It is important to acknowledge the limitations of this study. Firstly, the sample size was small in some cases, as in the case of infliximab and vedolizumab. This limitation stemmed from the fact that vedolizumab, consistent with local treatment protocols, is often reserved for patients who have failed or are intolerant to previous lines of treatment. As a result, the pool of eligible patients for inclusion in the vedolizumab arm was inherently smaller compared to other treatment arms where medications might be used earlier in the treatment algorithm. Additionally, in the case of infliximab, the small sample size can be attributed to logistical challenges associated with its intravenous administration and the logistical circuit involved. Despite the fact that the number of patients receiving infliximab treatment is considerably higher than that of vedolizumab, difficulties in recruitment persisted due to these logistical hurdles. Consequently, the smaller sample size in the infliximab and vedolizumab arms may have also affected this study’s statistical power and its ability to draw robust conclusions regarding these treatments.

Secondly, in the case of ADAs, the sample size was notably constrained due to the relatively low prevalence of immunogenicity within the patient population. This inherent scarcity of ADAs posed challenges in accruing a sufficiently large sample size for robust statistical analysis. Furthermore, the limited sample size hindered the feasibility of conducting a comprehensive comparison between techniques for ADA detection. Additionally, the differences in units of measurement between the assays employed for ADA detection further complicated the comparability of results. These discrepancies in measurement units hindered our ability to conduct a comprehensive and reliable comparison, thus limiting the interpretability and robustness of our findings pertaining to ADAs.

### 4.4. Future Research Directions

In light of the limitations encountered in our study, we have outlined several strategies for future research to address these challenges effectively. Firstly, expanding our sample size is paramount to bolstering the robustness and applicability of our findings. This expansion will notably involve including a broader cohort of patients with detectable concentrations of anti-drug antibodies (ADAs). Additionally, we intend to explore alternative methodologies or refine existing techniques to minimize potential biases and discrepancies observed in our results.

Moreover, longitudinal studies are warranted to investigate the long-term implications of different quantification techniques for treatment outcomes and patient prognosis. Furthermore, collaboration with other research groups and institutions may offer valuable insights and resources to overcome the limitations highlighted in our study. Overall, we are committed to advancing our understanding of biological drug quantification techniques and immunogenicity in patients with immune-mediated diseases through ongoing research efforts.

## 5. Conclusions

Despite some differences in the results, both techniques prove valuable in clinical practice for monitoring cADA and cINF. However, the results are modest for cUST and cVED, so caution is recommended and further research is needed to understand their clinical implications. In the case of ADA, it was not possible to assess the comparison. The adoption of CLIA warrants consideration in clinical practice, given its potential benefits for optimizing treatment outcomes in IMIDs, although the choice between techniques may depend on the particular characteristics of the drug being monitored. However, it is essential to acknowledge the limitations of our study, especially the small sample sizes.

## Figures and Tables

**Figure 1 biomedicines-12-00839-f001:**
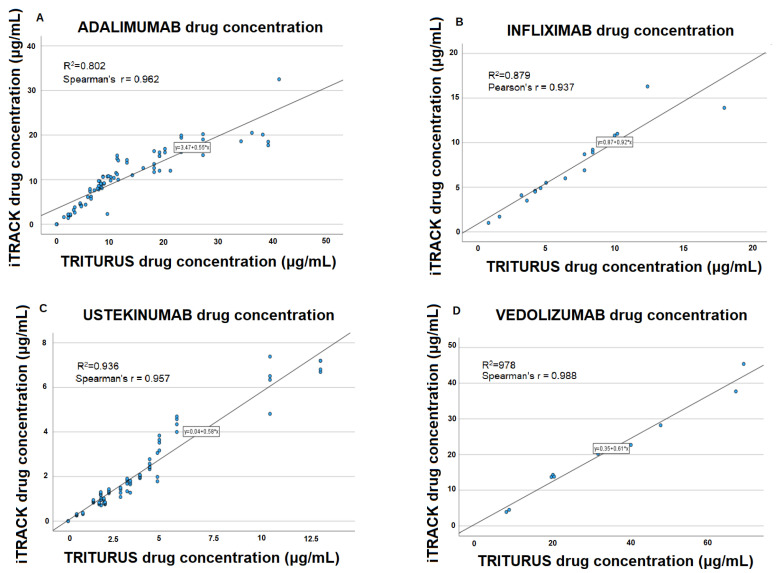
Lineal regression and correlation values for adalimumab drug concentration (**A**), infliximab drug concentration (**B**), ustekinumab drug concentration (**C**), and vedolizumab drug concentration (**D**) when they were measured using i-TRACK (CLIA) and TRITURUS (ELISA). R2 values were calculated for all linear regressions. Pearson’s test was used when the variables were normally distributed and Spearman’s test when they were not. CLIA: chemiluminescence immunoassay, ELISA: Enzyme-Linked ImmunoSorbent Assay.

**Figure 2 biomedicines-12-00839-f002:**
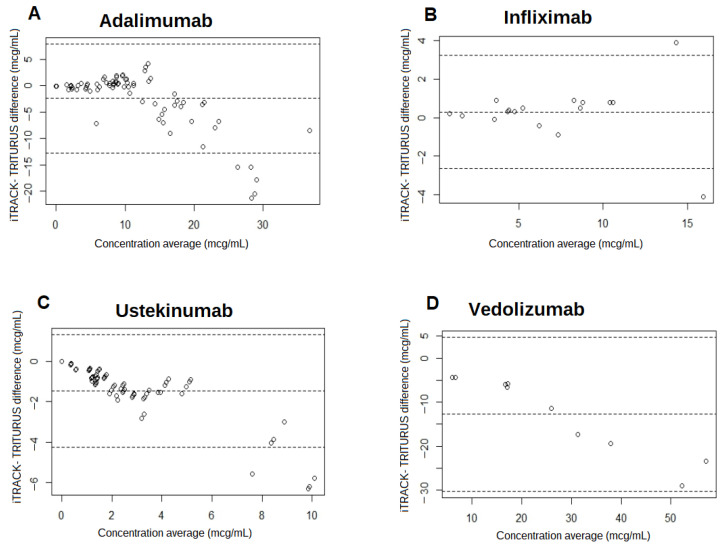
Bland–Altman plots comparing i-TRACK values with TRITURUS values for adalimumab drug concentration (**A**), infliximab drug concentration (**B**), ustekinumab drug concentration (**C**), and vedolizumab drug concentration (**D**). The y-axis represents the difference between the value of the two measures (i-TRACK and TRITURUS). The x-axis represents the average of the two measures (i-TRACK and TRITURUS). The dashed line in the center represents the mean difference limit or bias. The dashed lines at the ends represent the limits of agreement when using a 95% confidence interval. mcg: micrograms, mL: milliliters.

**Table 1 biomedicines-12-00839-t001:** Data agreement between TRITURUS and i-TRACK values for adalimumab, infliximab, vedolizumab and ustekinumab. Values are stratified according to the therapeutic ranges established by the literature for each drug. CI: Confidence interval; μg = micrograms; mL = milliliters.

**ADALIMUMAB**		**ITRACK Drug Concentration**			**Total**
Triturus drug concentration		<5 μg/mL	(5–8) μg/mL	>8 μg/mL	
	<5 μg/mL	15	0	0	15
	(5–8) μg/mL	1	8	3	12
	>8 μg/mL	1	0	47	48
Total		17	8	50	75
	Kappa value: 0.87 (CI: 95%: 0.76–0.98)				
**INFLIXIMAB**		**ITRACK drug concentration**			**Total**
Triturus drug concentration		<3 μg/mL	(3–7) μg/mL	>7 μg/mL	
	<3 μg/mL	2	0	0	2
	(3–7) μg/mL	0	7	0	7
	>7 μg/mL	0	1	7	8
Total		2	8	7	17
	Kappa value: 0.902 (CI: 95%: 0.715–1.089)				
**USTEKINUMAB**		**ITRACK drug concentration**			**Total**
Triturus drug concentration		<1 μg/mL	(1–4.5) μg/mL	>4.5 μg/mL	
	<1 μg/mL	9	0	0	9
	(1–4.5) μg/mL	17	31	0	48
	>4.5 μg/mL	0	9	10	19
Total		26	40	10	76
	Kappa value: 0.424 (CI: 95%: 0.245–0.604)				
**VEDOLIZUMAB**		**ITRACK drug concentration**		**Total**	
Triturus drug concentration		<25 μg/mL	>25 μg/mL		
	<25 μg/mL	5	0	5	
	>25 μg/mL	2	3	5	
Total		7	3	10	
	Kappa value: 0.424 (CI: 95%: –0.072–0.920)				

## Data Availability

Due to study participant privacy, data cannot be shared openly.

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
