# Peer review of "Observational Study to Compare Biological Drug Concentration Quantification Techniques and Immunogenicity in Patients with Immune-Mediated Diseases"

_biomedicines, 2024, doi:10.3390/biomedicines12040839_

Round 1

Reviewer 1 Report

Comments and Suggestions for Authors

Overall comments:

The subject matter of this manuscript deals with the comparison of biological drug concentration quantification techniques and immunogenicity in patients with immune-mediated diseases. The authors compare chemiluminescence immunoassays with ELISA for measuring trough levels and anti-drug antibodies of biological drugs. It was shown that while both techniques show excellent correlation, i-TRACK tends to yield lower values, particularly at higher concentrations.

Seemingly, the manuscript has been well-written with the sufficient research outputs. However, I think the innovation and importance of this manuscript are not enough to meet the requirements of the journal. I agree authors conducted a lot of experiments but I worry that this masks what appears to be very important subject. Thus, this paper seems difficult to secure its publication to this journal in the current state. Together with this major point, there are some specific concerns that should be clearly addressed.

Specific concerns:

1. In the Introduction, The authors should elucidate the rationale behind selecting the screening targets (infliximab, adalimumab, vedolizumab, and ustekinumab) based on their therapeutic importance.

2. Enlarge the axis labels and functions in the center of the graphs in Figure 1. To maintain consistency, employ abbreviations in the graph titles (as they were abbreviated in the manuscript) or place them next to the full form (for figures 1 and 2, table 1).

3. In section 4.3, offer the authors' perspectives on their future plans to address the limitations encountered in their work.

Comments on the Quality of English Language

none

Author Response

Comments and Suggestions for Authors

Overall comments:

The subject matter of this manuscript deals with the comparison of biological drug concentration quantification techniques and immunogenicity in patients with immune-mediated diseases. The authors compare chemiluminescence immunoassays with ELISA for measuring trough levels and anti-drug antibodies of biological drugs. It was shown that while both techniques show excellent correlation, i-TRACK tends to yield lower values, particularly at higher concentrations.

Seemingly, the manuscript has been well-written with the sufficient research outputs. However, I think the innovation and importance of this manuscript are not enough to meet the requirements of the journal. I agree authors conducted a lot of experiments but I worry that this masks what appears to be very important subject. Thus, this paper seems difficult to secure its publication to this journal in the current state. Together with this major point, there are some specific concerns that should be clearly addressed.

Dear Reviewer,

Thank you for taking the time to review our manuscript titled "Comparison of Biological Drug Concentration Quantification Techniques and Immunogenicity in Patients with Immune-Mediated Diseases." We appreciate your constructive feedback and have carefully considered your comments and suggestions.

Specific concerns:

1. In the Introduction, The authors should elucidate the rationale behind selecting the screening targets (infliximab, adalimumab, vedolizumab, and ustekinumab) based on their therapeutic importance.

We have provided a clearer rationale in the Introduction section (line 74) regarding the selection of screening targets, including infliximab, adalimumab, vedolizumab, and ustekinumab, based on their therapeutic importance. This helps readers understand the significance of our study objectives.

2. Enlarge the axis labels and functions in the center of the graphs in Figure 1. To maintain consistency, employ abbreviations in the graph titles (as they were abbreviated in the manuscript) or place them next to the full form (for figures 1 and 2, table 1).

We have ensured that the axis labels and functions in Figure 1 are enlarged for better readability. Additionally, we have removed abbreviations from the figures for their full form, as recommended.

3. In section 4.3, offer the authors' perspectives on their future plans to address the limitations encountered in their work.

We have created a new section 4.4 (line 371), where we have provided a discussion on our future plans to address the limitations encountered in our work. This includes outlining potential strategies for overcoming the challenges identified in our study, thereby enhancing the robustness and applicability of our findings.

We believe that addressing these specific concerns strengthens the manuscript and aligns it more closely with the standards and expectations of the journal. We are committed to making the necessary revisions to improve the quality and impact of our research.

Thank you once again for your valuable feedback, and we look forward to the opportunity to resubmit our revised manuscript for further consideration.

Best regards,

Alejandro Martínez

Reviewer 2 Report

Comments and Suggestions for Authors

The authors of this manuscript (Observational Study to Compare Biological 2 Drug Concentration Quantification Techniques 3 and Immunogenicity in Patients with 4 Immune-Mediated Diseases) have performed a careful and statistically significant comparison among different methods for the quantification of several drugs used for the treatment of immune diseases. I find that their findings will be very useful for the clinical field, where diagnostics can be sometimes very straightforward and complex, so more alternatives that may give good results to make clinical decisions are welcome. I recommend publication as it is, with no further changes.

Author Response

Comments and Suggestions for Authors

The authors of this manuscript (Observational Study to Compare Biological 2 Drug Concentration Quantification Techniques 3 and Immunogenicity in Patients with 4 Immune-Mediated Diseases) have performed a careful and statistically significant comparison among different methods for the quantification of several drugs used for the treatment of immune diseases. I find that their findings will be very useful for the clinical field, where diagnostics can be sometimes very straightforward and complex, so more alternatives that may give good results to make clinical decisions are welcome. I recommend publication as it is, with no further changes.

Dear Reviewer,

Thank you for your positive evaluation of our manuscript titled "Observational Study to Compare Biological Drug Concentration Quantification Techniques and Immunogenicity in Patients with Immune-Mediated Diseases." We sincerely appreciate your recognition of the significance of our findings for the clinical field.

Your encouragement to publish the manuscript as it is, without further changes, is greatly appreciated. We are pleased that you find our study's comparison of different quantification methods for immune disease treatment drugs to be valuable and relevant for clinical decision-making.

We will proceed with the publication process accordingly and look forward to contributing our research to the scientific community.

Thank you once again for your support and feedback.

Best regards,

Reviewer 3 Report

Comments and Suggestions for Authors

The manuscript by Martínez-Pradeda and co-authors describes the comparative statistical analysis of different methodologies for drug quantification in patients with immune-mediated diseases. Mainly, ELISA technique is compared with chemiluminescence immunoassay (CLIA), the latter being less time-consuming. The results are important. However, the manuscript has the following technical and methodological drawbacks.

Line 18:  Please replace "Mesuring" with "Measuring". Check also throughout the manuscript.

Line 23:  Please replace "chimiluminescent" with "chemiluminescent".

Line 26:  Please replace "asssays" with "assays".

Line 46:  Please replace "agents’s" with "agents’".

Line 120:  The phrase "similar to 0" looks strange.

Line 122:  If "0" and "missing" are categorical variables, then they should be explained more clearly.

Line 258:  The phrase "Untill our knowledge" looks strange.

More comments should be given concerning different sample size (number of data points) on the investigation of different components. For example, why the number of points for vedolizumab is about four times smaller than that for adalimumab (Figure 1)?

It is worth giving some rationale for underestimation of larger concentrations (especially for adalimumab) by iTRACK vs TRITURUS methodology.

Summarizing, I recommend major revision of the manuscript before acceptance.

Comments on the Quality of English Language

Misspellings are met in the manuscript. Some of them are mentioned in the review.

Author Response

Comments and Suggestions for Authors

The manuscript by Martínez-Pradeda and co-authors describes the comparative statistical analysis of different methodologies for drug quantification in patients with immune-mediated diseases. Mainly, ELISA technique is compared with chemiluminescence immunoassay (CLIA), the latter being less time-consuming. The results are important. However, the manuscript has the following technical and methodological drawbacks.

Dear Reviewer,

Thank you for your detailed review of our manuscript titled "Comparative Analysis of Drug Quantification Methodologies in Patients with Immune-Mediated Diseases." We appreciate your constructive feedback and have addressed the issues you raised.

Line 18: Please replace "Mesuring" with "Measuring". Check also throughout the manuscript.

Line 23: Please replace "chimiluminescent" with "chemiluminescent".

Line 26: Please replace "asssays" with "assays".

Line 46: Please replace "agents’s" with "agents’".

Line 120: The phrase "similar to 0" looks strange.

Line 122: If "0" and "missing" are categorical variables, then they should be explained more clearly.

Line 258: The phrase "Untill our knowledge" looks strange.

We have corrected all spelling errors throughout the manuscript. We appreciate your diligence in identifying these errors.

More comments should be given concerning different sample size (number of data points) on the investigation of different components. For example, why the number of points for vedolizumab is about four times smaller than that for adalimumab (Figure 1)?

Regarding the sample size discrepancy between vedolizumab and adalimumab data points in Figure 1, we acknowledge the disparity and have included additional rationale in the revised manuscript to address this discrepancy. We have included in the limitations section (line 347) a more extensive explanation of the causes of this different sample size, since we considered it to be the most appropriate place to refer to the entire manuscript.

It is worth giving some rationale for underestimation of larger concentrations (especially for adalimumab) by iTRACK vs TRITURUS methodology.

Furthermore, we have provided a detailed rationale for the observed underestimation of larger concentrations, particularly for adalimumab, by the iTRACK vs TRITURUS methodology (line 261). While we found limited explanation in the consulted literature regarding the differences in trends observed among the results of different methods, we have added the possible causes indicated by the different authors consulted, and emphasized the need to explore these causes further.

Summarizing, I recommend major revision of the manuscript before acceptance.

Comments on the Quality of English Language

Misspellings are met in the manuscript. Some of them are mentioned in the review.

We have thoroughly revised the manuscript to address these issues and enhance its clarity and scientific rigor. We believe these revisions significantly improve the quality of the manuscript.

Thank you for your valuable feedback, and we look forward to your further evaluation of the revised manuscript.

Sincerely,

Alejandro Martínez

Reviewer 4 Report

Comments and Suggestions for Authors

The paper explores the use of therapeutic drug monitoring (TDM) for biological agents in immune-mediated inflammatory diseases (IMIDs) like inflammatory bowel disease (IBD) and rheumatoid arthritis (RA). It focuses on comparing two techniques, ELISA and CLIA, for measuring drug concentrations and anti-drug antibodies (ADAs). While both methods show good correlation for drugs like adalimumab and infliximab, there are discrepancies for ustekinumab and vedolizumab, suggesting caution and further research are needed, especially for ADA quantification. CLIA shows promise for rapid decision-making in clinical practice. However, limitations include small sample sizes and differences in measurement units. Overall, the study underscores the importance of selecting the most appropriate technique tailored to each specific biologic drug for optimal treatment outcomes in IMIDs.

What implications do the findings of the study have for clinical practice, especially regarding the adoption of CLIA for therapeutic drug monitoring?

Comments on the Quality of English Language

Overall, the English language used in the paper is proficient and demonstrates a good understanding of scientific terminology and concepts. The sentences are generally well-structured, and technical terms are used appropriately. However, there are some instances where sentence structure could be improved for better clarity, and there are occasional grammatical errors and awkward phrasings. Additionally, the use of punctuation, particularly commas and semicolons, could be more consistent to enhance readability. Overall, the quality of English language is satisfactory but could benefit from minor revisions for improved clarity and coherence.

Author Response

Comments and Suggestions for Authors

The paper explores the use of therapeutic drug monitoring (TDM) for biological agents in immune-mediated inflammatory diseases (IMIDs) like inflammatory bowel disease (IBD) and rheumatoid arthritis (RA). It focuses on comparing two techniques, ELISA and CLIA, for measuring drug concentrations and anti-drug antibodies (ADAs). While both methods show good correlation for drugs like adalimumab and infliximab, there are discrepancies for ustekinumab and vedolizumab, suggesting caution and further research are needed, especially for ADA quantification. CLIA shows promise for rapid decision-making in clinical practice. However, limitations include small sample sizes and differences in measurement units. Overall, the study underscores the importance of selecting the most appropriate technique tailored to each specific biologic drug for optimal treatment outcomes in IMIDs.

What implications do the findings of the study have for clinical practice, especially regarding the adoption of CLIA for therapeutic drug monitoring?

Dear Reviewer,

Thank you for your valuable feedback on our manuscript, "Comparative Analysis of Drug Quantification Methodologies in Patients with Immune-Mediated Diseases." Your insights have been invaluable in refining our research.

Our study investigates therapeutic drug monitoring (TDM) for biological agents in immune-mediated inflammatory diseases (IMIDs), comparing ELISA and CLIA techniques. While both methods show good correlation for some drugs, disparities exist for others, highlighting the need for caution and further research, especially in ADA quantification. Despite challenges, CLIA holds promise for rapid decision-making in clinical practice.

Regarding the implications for clinical practice, our findings underscore the importance of selecting the most appropriate technique tailored to each specific biologic drug. CLIA's potential benefits, including expeditious results and comparable performance with ELISA, advocate for its consideration as a valuable tool in routine patient care. We have elaborated on the practical application of these findings in both the discussion and the conclusion sections, providing a more detailed explanation (lines 333 and 391).

Comments on the Quality of English Language

Overall, the English language used in the paper is proficient and demonstrates a good understanding of scientific terminology and concepts. The sentences are generally well-structured, and technical terms are used appropriately. However, there are some instances where sentence structure could be improved for better clarity, and there are occasional grammatical errors and awkward phrasings. Additionally, the use of punctuation, particularly commas and semicolons, could be more consistent to enhance readability. Overall, the quality of English language is satisfactory but could benefit from minor revisions for improved clarity and coherence.

We appreciate your comments on language quality and will ensure improved clarity, coherence, and consistency in the revised manuscript.

Thank you for your continued evaluation of our work.

Best regards,

Alejandro Martinez

Round 2

Reviewer 1 Report

Comments and Suggestions for Authors

The authors have provided suboptimal, albeit neither perfect nor the best, replies to some critical issues raised in the previous review stage.

It is considered that the present version of the manuscript was revised to some extent according to all the reviewers' comments.

So, this manuscript would be acceptable, unless otherwise decided by other reviewers.

Reviewer 3 Report

Comments and Suggestions for Authors

The revised version of the manuscript has been improved significantly by the authors. I recommend acceptance of the manuscript for publication in the revised form.